# Study on the Dynamic Changes in Non-Volatile Metabolites of Rizhao Green Tea Based on Metabolomics

**DOI:** 10.3390/molecules28217447

**Published:** 2023-11-06

**Authors:** Ao Sun, Guolong Liu, Luyan Sun, Chun Li, Qiu Wu, Jianhua Gao, Yuanzhi Xia, Yue Geng

**Affiliations:** 1Key Laboratory of Food Nutrition and Safety of SDNU, Provincial Key Laboratory of Animal Resistant Biology, College of Life Science, Shandong Normal University, Jinan 250014, China; 17853136957@163.com (A.S.); 18006402224@163.com (G.L.); skz1137081406@163.com (L.S.); 2020020893@stu.sdnu.edu.cn (C.L.); wuqiu@sdnu.edu.cn (Q.W.); 2Shandong Rizhao Shenggushan Tea Farm Co., Ltd., Rizhao 276827, China; 3Jinan Three Thousand Tea Grower Co., Ltd., Jinan 250022, China

**Keywords:** taste-related substances, Rizhao green tea, UHPLC-Q Exactive MS, non-targeted metabolomics, processing

## Abstract

The processing of tea leaves plays a crucial role in the formation of the taste of the resulting tea. In order to study the compositions of and changes in taste-related substances during the processing of Rizhao green tea, non-targeted metabolomics was used, based on UHPLC-Q Exactive MS. Totals of 529, 349, and 206 non-volatile metabolites were identified using three different detection modes, of which 112 secondary metabolites were significantly changed. Significant variations in secondary metabolites were observed during processing, especially during the drying stage, and the conversion intensity levels of non-volatile metabolites were consistent with the law of “Drying > Fixation > Rolling”. The DOT method was used to screen tea-quality-related compounds that contributed significantly to the taste of Rizhao green tea, including (−)-epicatechin gallate, (−)-epicatechin gallate, gallic acid, L-theanine, and L-leucine, which make important contributions to taste profiles, such as umami and bitterness. Metabolic pathway analysis revealed that purine metabolism, caffeine metabolism, and tyrosine metabolism perform key roles in the processing of Rizhao green tea in different processing stages. The results of this study provide a theoretical basis for tea processing and practical advice for the food industry.

## 1. Introduction

Tea is one of the most widely consumed beverages worldwide. Depending on different processing methods, it can be classified into six categories: green, white, yellow, oolong, black, and dark tea. The chemical and sensory characteristics of tea vary significantly among different types of tea [1]. Green tea, a non-fermented tea, accounts for 20% of global tea production and is most commonly consumed in East Asia. However, due to its extensive biological functions and health properties, its popularity is increasing in other parts of the world, such as Europe and the United States [2]. Previous studies have shown that the processing of tea has certain impacts on secondary metabolites, such as theanine, catechins, and caffeine; for example, drying is one of the most important processes involved in making green tea [3], and high-temperature treatment (≥180 °C) can change the configuration of chiral carbon, thus reducing the contents of cis-catechins and increasing the content of trans-catechins [4].

Rizhao green tea is produced in Rizhao City, Shandong Province, in China. It is made from the buds, fresh leaves, and tender stems of tea trees that grow in a specific range under natural ecological conditions, and it is processed through fixation, rolling, and drying techniques [5]. In 2006, Rizhao green tea was evaluated as a geographically indicated agricultural product of Rizhao, with the highest brand value in Shandong Province. And, in 2020, it was successfully included in the EU-certified China Geographical Indication Protection Catalogue [6]. Compared with southern tea, the larger DIF (difference between day and night temperature) in Rizhao benefits the growth of tea trees and the accumulation of chemical components related to tea quality, such as tea polyphenols, amino acids, and beneficial minerals, giving Rizhao green tea characteristics of tender green leaf bottoms, a bright broth color, a rich chestnut flavor, and a refreshing taste. Currently, there is insufficient basic research on the quality and flavor profile changes in Rizhao green tea during its processing.

Taste is one of the important sensory indicators for evaluating the flavor quality of green tea. The taste of green tea is mostly influenced by non-volatile components, with its qualities mainly including astringency, bitterness, umami, and sweetness. Among them, bitterness and astringency are two important quality attributes in green tea tasting [7]. Most bitter substances are accompanied by astringency. Currently, the main bitter and astringent substances identified in green tea can be roughly divided into two categories: polyphenols and alkaloids. Yu et al. [8] determined that caffeine and Epigallocatechin gallate (EGCG) were the main contributors to the bitterness and astringency of tea broth, respectively, through taste reconstruction experiments and by establishing regression models. The main components of the umami flavor in tea broth are theanine, glutamic acid, and aspartic acid. T1R1 (Taste receptor type 1 member 1) and T1R3 (Taste receptor type 1 member 3) are the umami receptors, and their transmission channels are consistent with the sweet taste transmission channels [9]. Previous research has mainly focused on the detection of the total amounts of these substances or the changes in a single substance during processing, usually characterizing a group or several groups of compounds, resulting in incomplete results [10]. In recent years, metabolomics technology has been widely used in the field of tea research, due to its advantages of fast separation, high sensitivity, and wide coverage range, which can comprehensively characterize the non-volatile components in tea. For example, based on UPLC-Q-TOF/MS and chemometrics, the effects of different cultivars, planting elevations, and processing methods on the metabolites of Xinyang Maojian tea were studied [11]. The study showed that 54 metabolites, such as catechins and amino acids, were related to the differentiation of four tea tree cultivars, and 27 metabolites, such as theanine, were closely related to the altitude, while the metabolite differences between hand-made tea and machine-made tea were not significant. Wang et al. [12] detected the changes in metabolites related to the advantages of tea tree hybrids through LC-ESI-MS/MS and found that nucleotides, alkaloids, organic acids, and tannins showed positive hybrid advantages, while catechins and free amino acids showed negative hybrid advantages. Metabolomics has become a powerful tool for exploring the dynamic changes in non-volatile metabolites in tea with respect to different cultivars, regions, storage environments, and processing techniques. In addition, dose-over-threshold (DOT) refers to the ratio of the concentration of a compound to the taste threshold, which can be used to determine the contribution of non-volatile substances to the taste characteristics of samples. Generally, compounds with a DOT > 1 have an important impact on the taste of tea broth [13]. Lisa Ullrich et al. [14] explored the contributions of compounds in chocolate to sensory attributes, such as cocoa flavor, roasted flavor, and astringency, by calculating their DOT values. This method has been applied to food fields such as alcohol [15] but has been less applied in the determination of the taste characteristics of Rizhao green tea.

To investigate the dynamic changes in non-volatile metabolites during the processing of Rizhao green tea and the effects of processing on tea flavor substances, this study used non-targeted metabolomics technology, based on UHPLC-Q Exactive MS, to analyze the metabolites of Rizhao green tea at different processing stages. Meanwhile, various statistical analysis methods, such as principal component analysis (PCA), partial least squares discriminant analysis (PLS-DA), and orthogonal partial least squares discriminant analysis (OPLS-DA), were used to clarify the dynamic evolution of tea metabolites. In addition, differential metabolites between adjacent processing stages were screened, their contributions to the sample taste were determined using the DOT method, and the effects of processing on these metabolites were explored. This study is expected to provide a theoretical reference and an objective basis for the formation mechanism of taste substances in high-quality green tea.

## 2. Results and Discussion

### 2.1. Non-Targeted Metabolomics Analysis of Rizhao Green Tea Samples

Figure 1A and Appendix A show the total ion chromatograms (TICs) obtained from Rizhao tea samples at different processing stages using a semi-polar C18 chromatographic column in different ion modes. A total of 529 and 349 metabolite ion features were detected and qualitatively identified in positive and negative ionization, respectively, as shown in Appendix A. Most TIC peaks could be found in both ESI+ and ESI− modes. Therefore, the ESI+ mode was selected for data collection. In addition, due to the dietary habit of brewing tea with boiling water, a mixed solution of water and methanol was designed as the solvent for extracting tea components, in order to increase the extraction rate of polar components in tea [16]. Appendix A shows the TIC obtained from a Hilic column in ESI+, and a total of 206 metabolite ion features were detected and identified (Appendix A). Thus, the Hilic column, as a polar chromatographic column, was used to complement the data collected by the C18 column. The results of the C18 column in ESI+ are presented in the main text, while the data and results for the other two detection methods can be found in the supplementary materials.

Figure 1B shows the TIC of QC samples (n = 6), and Figure 1C is a 3D-PCA plot of QC samples. The TIC baseline is relatively stable, and the response intensity and retention time of each peak overlap. The good QC sample clustering in PCA indicates that the experimental data recorded in this study have good repeatability and reliability.

### 2.2. Multivariate Statistical Analysis

The data collected by high-throughput instruments were transformed into visual forms through multivariate statistical analysis to reveal the similarities and differences between samples. Therefore, in order to compare the differences in metabolite profiles of Rizhao green tea at different processing stages, PCA, PLS-DA, and OPLS-DA models were analyzed for qualitative substances, where 529 non-volatile metabolites were used as independent variables and four treatment groups were used as dependent variables. As shown in Figure 2A, in the PCA model containing QC samples, PC1 and PC2 explained 35.5% and 14.1% of the total variance, respectively, and the sample metabolite profiles were able to completely classify and identify samples from four different processing stages, indicating that processing had a significant impact on tea metabolome. In addition, the overlap of QC was high, indicating good reproducibility of the experiment.

PLS-DA (Figure 2B) and OPLS-DA (Figure 2C) are consistent with the principal component analysis results in that the samples at each stage are separated to varying degrees. In the direction of PC1, the samples can be clearly divided into two categories based on the order of tea processing: samples before drying (TLs, FLs, and RLs) and samples after drying (GT). In the direction of PC2, the first category of tea samples can be further divided into two subgroups: samples before fixation (TLs) and samples after fixation (FLs and RLs). Compared with other samples, the separation between tea leaves (TLs) and green tea (GT) is higher, while the separation between fixated leaves (FLs) and rolled leaves (RLs) is difficult, which may be due to the small temperature change after fixation and rolling and the absence of enzymatic reactions [17], thus leading to the insignificant changes in substance at this stage. These results indicate that the drying stage is the most critical processing stage for non-volatile substance changes, followed by fixation, while the influence of rolling is minimal, which is consistent with the results of ESI− (Appendix A) and the Hilic column (Appendix A). In the confidence test, Q^2^ measures the predictive ability of the model, while R^2^ determines the goodness of fit. The closer R^2^ and Q^2^ are to 1, the more reliable the model [18]. Figure 2D shows the 200-times confidence test, indicating that the results of the multivariate statistical analysis are reliable.

### 2.3. Screening of Differential Metabolites

Based on the changes in *p*-values and fold changes (FCs), potential characteristic metabolites between adjacent processing stages were screened, and the results are shown in a figure. In the volcano plot (Figure 3), the log value of the FC is plotted on the horizontal axis, and the negative logarithm value of the *p*-value is plotted on the vertical axis. With the screening criteria of FC > 2 or FC < 0.5 and *p* < 0.05, potential metabolites with significant differences in tea samples from different processing stages were preliminarily screened. Red dots represent up-regulated metabolites with FC > 2 and *p* < 0.05, green dots represent down-regulated metabolites with FC < 0.5 and *p* < 0.05, and gray dots represent non-differential metabolites. According to the univariate statistical analysis, totals of 212, 9, and 221 differential metabolites were identified for TLs vs. FLs, FLs vs. RLs, and RLs vs. GT, respectively.

To more accurately screen differential metabolites, pairwise OPLS-DA and VIP-value calculation were performed on the metabolites of adjacent processing stages of tea samples (Appendix A). Based on VIP > 1, 48 differential metabolites were further selected from the aforementioned differential metabolites, and their distribution in each processing stage is shown in Figure 3D. There were 20 and 19 types of metabolites affected by fixation and drying temperature, respectively, providing further evidence of the vital role of fixation and drying in converting non-volatile metabolites during tea processing. There were common differential metabolites among different control groups, as well as unique ones. In addition, using the same screening method and criteria, 47 and 29 differential metabolites were screened in the C18 column (ESI−) and Hilic column (ESI+), respectively, and their distribution in different processing stages is detailed in Appendix A.

### 2.4. Changes in Taste-Related Substances

A total of 112 differential substances were screened by the C18 column (ESI+/−) and Hilic column (ESI+), including 11 amino acids and their derivatives, 10 nucleotides and their derivatives, 16 flavonoids, 10 phenolic acids, 15 lipids, and 16 organic acids, as well as 34 other metabolites including terpenes, alkaloids, coumarins, and sugars, with repeated substances using data from the C18 column (ESI+). To intuitively show the dynamic changes in differential metabolites during processing, their relative contents were plotted as a heatmap (Figure 4). The color ranges from red to blue, representing the relative content of metabolites from high to low.

Not all differential metabolites are related to tea flavor, and different substances contribute differently to flavor characteristics. Therefore, PCA was performed on the differential metabolites screened by different detection methods using SPSS 27.0, and their initial eigenvalues and variance contribution rates are shown in Appendix A. According to the table, the extracted substances detected by each method were finally determined to have two principal components, with variance contribution rates of 92.893%, 94.976%, and 96.862%, which essentially explain most of the information of the variables. Based on the absolute values of the initial eigenvalues of each substance, the first principal component includes the total information of substances such as picolinic acid, quercetin, myricetin, gallic acid, procyanidin B2, L-theanine, L-leucine, and choline O-sulfate; the second principal component includes the total information of substances such as L-norleucine, allysine, ECG, EGCG, methyl gallate, salicylic acid, adenine, and 1,3,5-norcaratriene.

With thresholds determined from published research and relative concentrations obtained using the peak area normalization method, the DOT values of these substances were calculated, as shown in Table 1. Metabolites with DOT > 1 were selected as contributors to the taste, and their relative content changes at different processing stages were focused on in Figure 5. Among them, four compounds, namely EGCG (DOT: 1.763), gallic acid (DOT: 1.397), ECG (DOT: 1.347), and L-theanine (DOT: 2.268), were the main contributors to the acidity, bitterness, and umami in Rizhao green tea water.

Using published research, substances related to tea quality characteristics were selected, and the relative content changes of these substances at different processing stages were the focus of the study (Figure 5). The astringent substances in tea are mainly polyphenols, of which the flavanols are the highest in content, accounting for about 80% of the total polyphenols, with catechins being the main representatives [19]. ECG and EGCG are ester-type catechins that are related to the astringency of tea. As one of the most unstable catechins, EGCG can undergo physical transformation by complexing with salivary mucin, as well as chemical transformation through neutral-pH-mediated hydrolysis and dimerization, affecting flavor perception [20]. Procyanidin is a class of oligomeric flavonoids composed of (+)-catechin and (−)-epicatechin and is bitter. In this study, procyanidin B2 significantly increased in the drying stage, which may be due to the fact that higher temperatures can catalyze the degradation of procyanidin C1 to generate procyanidin B2 [21]. Moreover, the strong mechanical stress applied during the rolling stage damages the internal tissue cell structure of the tea leaves, increases the cell rupture rate [22], and allows the catechin substrate in the chloroplasts and vacuoles and the catechin oxidase in the cytoplasm to come into contact [23], which is conducive to the formation of dimeric catechins such as procyanidin. Flavonols and their derivatives are also important bioactive substances in tea, accounting for about 3–4% of the total dried weight of tea, and make great contributions to the flavor, taste, and color of tea infusions [24]. The experiment showed that the relative content of quercetin and myricetin showed an increasing trend during the fixation and drying stages, which is consistent with the results reported in [25]. The temperature increase during fixation and drying leads to enzyme hydrolysis, thereby increasing the content of flavonoids such as quercetin [26]. The increase in these substances will increase the bitterness and astringency of Rizhao green tea infusions.

The bitterness of a tea infusion mainly comes from alkaloids, tea polyphenols, and bitter amino acids. Among them, alkaloids in tea mainly include purine alkaloids, such as caffeine, theobromine, and theophylline [27]. Influenced by tea tree cultivars and processing methods, the content of caffeine in black tea is higher than that in green tea. Some studies have shown that the content of caffeine in black tea increased significantly by 69.4 times after processing [28], which may be the result of microbial fermentation. In this study, theophylline increased at the fixation stage and then gradually decreased, causing a decrease in the bitterness of the tea infusion. The lower theophylline content in Rizhao green tea contributes to its good sensory experience, which is also found in Enshi green tea [29]. Tea infusions contain various free amino acids, which exhibit different taste attributes; for example, leucine exhibits bitterness [30]. It has been found that compared to alkaloids, the concentration of bitter amino acids in a black tea infusion is relatively low and does not exceed the corresponding bitterness threshold. Meanwhile, sensory panel members could not perceive any differences in the intensity of taste quality after removing bitter amino acids, indicating that the contribution of bitter amino acids to tea broth bitterness is relatively small [31].

As the main free amino acid in tea, L-theanine contributes to the umami taste of tea and accounts for more than 50% of the total free amino acids, with the highest content in green tea [32,33]. During the fixation and drying stages, the thermal treatment deactivates enzyme activity and converts amino acids into aromatic compounds through the Maillard reaction, which explains the sharp decrease in the relative content of most amino acids during these two stages. N-ethyl-2-pyrrolidinone-substituted flavan-3-ols (EPSFs) are flavan-3-ol derivatives that are typically substituted at the C-6 and/or C-8 positions of flavanols by N-ethyl-2-pyrrolidinone, which is a product of the Strecker degradation of theanine [34]. Therefore, EPSFs are generally considered as adducts of theanine and flavanols. Peng et al. [35] confirmed that L-theanine and flavanols can react to form EPSFs under dry tea conditions, and EPSFs increase during tea baking, accompanied by a decrease in L-theanine, indicating that Strecker degradation and interaction with flavanols are important pathways for the degradation of green tea amino acids during the drying process. According to the different taste characteristics of amino acids, they can be classified into sweet, delicate, bitter, and sour amino acids. Leucine and proline are related to the bitter and sweet taste of brewed tea, respectively [36], and their relative content decreases compared to TLs.

Some nucleotides and their derivatives also have an umami taste. The content of nucleotides in tea is relatively low, and most of them can combine with taste receptors to significantly enhance the taste of tea infusions [37]. However, in this study, the contents of hypoxanthine, 1-methylxanthine, and adenosine were significantly reduced because nucleotides are metabolized through purine and pyrimidine degradation pathways by nonspecific nucleotidases and phosphatases, which are generally present in the vacuole. Due to the fermentation process, various enzymes were released from the vacuole, leading to a more significant decrease in nucleotide content in black tea compared to green tea [38].

Previous studies have shown that phenolic acids and organic acids are positively correlated with the smoothness, sweetness, thickness, and acidity of tea [39]. The present study showed that the relative content of GA significantly increased after high-temperature drying treatment, and similar changes were observed during the roasting of different aromatic oolong teas [35] and Qingzhuan tea [40], indicating strong oxidation–reduction and hydrolysis reactions during fixation and drying that promote the formation of phenolic acids [41].

**Table 1 molecules-28-07447-t001:** Relative content, thresholds, and DOT values of taste-related compounds.

No.	Name	Taste	Relative Content (%) ^a^				Taste Threshold (mg/L) ^b^	DOT
			TLs	FLs	RLs	GT		
1	Pipecolic acid	/	10.55050 ± 0.73458	3.69746 ± 0.72721	2.96563 ± 0.26097	0.78934 ± 0.07104	/	/
2	L-Norleucine	/	1.73371 ± 0.29184	0.98388 ± 0.15684	1.24953 ± 0.21237	1.45375 ± 0.13938	/	/
3	Allysine	/	0.80972 ± 0.35580	0.20860 ± 0.04917	0.16443 ± 0.02796	0.02419 ± 0.00186	/	
4	Quercetin	bitterness	0.00010 ± 0.00002	0.02025 ± 0.01095	0.06064 ± 0.02886	0.39393 ± 0.05495	/	/
5	Myricetin	bitterness	0.00004 ± 0.00001	0.00112 ± 0.00092	0.00223 ± 0.00142	0.16603 ± 0.01122	/	/
6	ECG	bitterness and astringency	0.00016 ± 0.00004	0.00120 ± 0.00079	0.02048 ± 0.01682	1.35739 ± 0.08095	201.5	1.347
7	EGCG	bitterness and astringency	0.00013 ± 0.00002	0.00037 ± 0.00010	0.00040 ± 0.00009	1.59685 ± 0.19673	181.2	1.763
8	Gallic acid	sourness and astringency	0.00071 ± 0.00011	0.00737 ± 0.00222	0.01160 ± 0.00431	0.23751 ± 0.08221	34	1.397
9	Procyanidin B2	bitterness	0.00018 ± 0.00007	0.00729 ± 0.00530	0.01620 ± 0.00694	0.10206 ± 0.00915	/	/
10	L-Theanine	umami	29.5388 ± 2.9869	19.4945 ± 4.4637	22.4670 ± 1.9170	11.8527 ± 1.3348	1045.2	2.268
11	Methyl gallate	/	0.0019 ± 0.0003	2.4876 ± 0.7255	2.4130 ± 0.5399	8.2612 ± 0.6417	/	/
12	Salicylic acid	/	0.0101 ± 0.0010	2.4634 ± 0.7006	2.4150 ± 0.5389	8.2178 ± 0.6353	/	/
13	L-Leucine	bitterness	13.94649 ± 2.18217	6.69525 ± 1.24796	5.63074 ± 1.01828	0.37763 ± 0.03533	1574.0	0.048
14	Adenine	umami	21.74214 ± 3.27621	27.66233 ± 0.88956	32.67142 ± 1.77501	1.90279 ± 0.08703	/	/
15	Choline O-Sulfate	/	6.24440 ± 1.31600	11.12400 ± 1.58627	8.56549 ± 0.79084	0.64120 ± 0.10662	/	/
16	1,3,5-Norcaratriene	/	9.75277 ± 0.8181	20.00826 ± 2.13235	18.77298 ± 2.64485	55.61301 ± 1.75427	/	/

^a^ Relative contents of compounds in samples are represented as mean ± standard error of mean (mean ± SEM); ^b^ “/” represents the threshold values of compounds that were not measured in water. All taste thresholds were obtained from published research [42,43].

### 2.5. Metabolic Pathway Analysis

The KEGG was used to perform pathway enrichment analysis on differential metabolites during the processing stages of Rizhao green tea, in order to obtain the distribution of differential metabolites in different metabolic and biosynthetic pathways. Figure 6A,C,E show the pathway enrichment diagram, where the size of a dot represents the number of related metabolites in a specific pathway, and the color represents the *p*-value. The magnitude of the *p*-value can reflect the importance of each pathway, with pathways where *p* < 0.05 being considered important. To visually compare the *p*-values of each pathway, a histogram (Figure 6B,D,F) was plotted with −lg (*p*-value) as the y-axis. It can be seen that metabolic and biosynthetic pathways are significantly enriched during the processing, including purine metabolism, tyrosine metabolism, caffeine metabolism, cysteine and methionine metabolism, and lysine degradation, of which purine metabolism and tyrosine metabolism were more important for the differential metabolites in different processing stages of the C18 column (ESI+). In the C18 column (ESI−) mode, glycerol phosphate shuttle, riboflavin metabolism, and glycerolipid metabolism were important factors contributing to the differences in metabolic profiles. Wang et al. [44] confirmed that there are differences in amino acid content and metabolism among tea cultivars and regions, which may be related to temperature, altitude, and precipitation. In addition, valine, leucine and isoleucine biosynthesis, nicotinate and nicotinamide metabolism, purine metabolism, and caffeine metabolism were important factors for the differences in sensory quality between samples at different stages of tea processing, and the results corresponded to a decrease in the content of purine analogs. The differential gene expression in the caffeine metabolism pathway corresponds to increased accumulation of caffeine, thereby achieving better sensory quality [45].

## 3. Materials and Methods

### 3.1. Chemicals and Instrument

Liquid chromatography–mass spectrometry (LC-MS)-grade acetonitrile was obtained from Honeywell (Morristown, NJ, USA), methanol of LC-MS grade was obtained from Merck (Darmstadt, Germany), formic acid of LC-MS grade was obtained from Thermo Fisher (purity > 98.0%) (Waltham, MA, USA), and analytically pure ammonium acetate was purchased from Sinopharm (Beijing, China). Deionized water was produced using a Milli-Q IQ 7000 water purification system (Merck, Darmstadt, Germany). Tea samples were ultrasonically extracted using an SB-800 (Scientz, Ningbo, China) and centrifuged using a SIGMA 3-18 K (Sigma-Aldrich, Saint Louis, MO, USA). The separation was completed on an Ultimate 3000-Q Exactive High-Performance Liquid Chromatography (UHPLC) system (Thermo Fisher, Waltham, MA, USA) with an ACQUITY UPLC BEH C18 column (2.1 × 50 mm, 1.7 μm) and a Hilic column (2.1 × 100 mm, 1.7 μm).

### 3.2. Tea Processing and Sampling

Rizhao green tea samples were collected on the first week of May 2021 at Shenggushan Tea Farm Co., Ltd. (Rizhao, Shandong, China). The fresh tea leaves with one or two leaves and one bud were repeatedly collected and screened by tea makers to ensure the samples were more representative in appearance, color, shape, and other aspects, and then the leaves were converted into commercial premium green tea of first grade, according to the standard process of withering, fixation, rolling, and drying. Postharvested samples were spread out from 4 h to 6 h. Following this step, tea samples were fixed at 230 °C for 1 min to reach a relative humidity of 60%, rolled at room temperature for 35 min, and dried at 90 °C for 20 min to produce Rizhao green tea. Tea samples (n = 6) from four key processing stages were named tea leaves (TLs), fixated leaves (FLs), rolled leaves (RLs), and green tea (GT) and were placed in centrifuge tubes, packed with sealing film, and transported to the laboratory at low temperature. The tea samples from each processing step were crushed with a household blender for 7–8 s and stored at −20 °C.

### 3.3. Extraction of Rizhao Green Tea Metabolites

Sample preparation for metabolomics analysis was performed according to our previous work [46] with slight modifications. First, 0.05 g of samples from different processing stages were crushed in a grinding dish and mixed with 15 mL of 70% methanol–water solution as an extraction solvent. An ultrasonic procedure at 40 KHz for 30 min under room temperature was performed to extract metabolites. The mixed solution was centrifuged at 12,000 rpm for 10 min at 4 °C. The supernatant was filtered with a 0.22 μm membrane (ANPEL, Shanghai, China) into labeled autosampler vials using disposable syringes (Shinva ANDE, Zibo, China) and stored at 4 °C for later analysis. Each sample was prepared at least 6 times. After the preparation of all extracts, 100 μL of each sample was mixed as a quality control (QC) sample to test the stability and reproducibility of UHPLC-Q Exactive MS.

### 3.4. Non-Targeted Metabolomics Analysis by UHPLC-Q Exactive MS

A UHPLC-Q Exactive MS instrument was used for the metabolomics analysis. Elution with a binary mobile phase was carried out at a flow rate of 0.4 mL/min. The column temperature was set at 40 °C and 30 °C, and the injection volume was 4 μL and 3 μL. The mobile phases A and B were optimized for two different gradients. For the C18 column, phase A was 0.1% FA water, and phase B was acetonitrile (0.1% FA). The gradient elution method was set as follows: 0–2 min, 0–10% phase B; 2–3 min, maintained at 10% phase B; 3–4 min, 10–15% phase B; 4–7 min, maintained at 15% phase B; 7–10 min, 15-40% phase B; 10–11 min, 40–55% phase B; 11–11.5 min, 55–100% phase B; 11.5–14.5 min, maintained at 100% phase B; 14.5–15 min, 100–0% phase B; 15–17 min, maintained at 0% phase B. For the Hilic column, phase A was 10 mM acetic acid water, and phase B was acetonitrile. The chromatographic gradient elution procedure was as follows: 0–4 min, 96% phase B; 4–10 min, 96–60% phase B; 10–12 min, maintained at 60% phase B; 12–13 min, 60–96% phase B; 13–15 min, maintained at 96% phase B.

The mass spectrometer used a dual jet stream electrospray ionization (ESI) source, and the positive ionization (ESI+) and negative ionization (ESI−) source conditions were located as follows: scan ranges of 75–1050 Da (+) and 80–1200 Da (−), spray voltages of 3.5 kV (+) and 3.1 kV (−); capillary temperature of 320 °C, sheath gas flow rate of 40 psi; aux gas flow rate of 10 psi. Complete data acquisition was performed using the FULL MS/DD-MS2 (TOP5) mode. The full MS resolution was 70,000 FWHM, the AGC target was 5 × 10^6^, and the maximum injection time was 200 ms; MS/MS resolution was 17,500 FWHM, and the maximum injection time was 25 ms.

### 3.5. Data Processing and Multivariate Statistical Analysis

The raw spectra obtained by UHPLC-Q Exactive MS were obtained using Xcalibar (Thermo Fisher Scientific, Waltham, MA, USA). The original data were subjected to peak alignment, retention time correction, and peak area extraction using the Compound Discoverer 3.2 program. Information on ion fragmentation of unknowns was obtained. Qualitative metabolites were identified by retrieving the MZcloud and Chemspider databases and validated using Chemspider (http://www.chemspider.com accessed on 15 January 2021). SIMCA14.1 software (UMetrics AB, Umea, Sweden) was used to perform multidimensional statistical analysis, including unsupervised principal component analysis (PCA), partial least squares discriminant analysis (PLS-DA), and orthogonal partial least squares discriminant analysis (OPLS-DA). To examine the quality of the models and to prevent overfitting of the models, cross-validation was performed on models. After multivariate statistical analysis, the differences between groups were tested with the Student *t*-test (*p* < 0.05), univariate statistical analysis was performed by one-way ANOVA using GraphPad Prism 8.0.1, and a volcano plot was used to screen differential metabolites between different processing stages. In this experiment, we chose *p* < 0.05, fold change (FC) > 2 or FC < 0.05, and VIP > 1 as criteria for significantly different metabolites. Principal component analysis of the differential metabolites was performed using IBM SPSS Statistics 27.0 to obtain the components contributing to the taste of Rizhao green tea. The relative concentration of differential metabolites was calculated by the peak area normalization method, and then the DOT value (relative concentration/threshold) of differential metabolites in GT samples was calculated; the final result was presented as the average of six datasets. In the end, the KEGG pathway database was used to perform a channel enrichment analysis of the differential metabolites in Rizhao green tea samples.

## 4. Conclusions

In this study, non-targeted metabolomics technology based on UHPLC-Q Exactive MS was used to identify and analyze the metabolites of Rizhao green tea at different processing stages. The results showed that there were differences in metabolites among different processing stages, with the drying stage being the most critical stage of metabolite changes, followed by the fixation stage, while the effect of rolling was minimal. Lipids were more abundant in TLs, while the contents of theanine, tannins, and flavonoids were higher in GT. Phenolic acids, coumarins, and terpenes were more abundant in FLs and RLs. The processing technology influenced the content of secondary metabolites in Rizhao green tea, while the content changes of metabolites such as catechins, amino acids, and phenolic acids affected the taste of Rizhao green tea. The astringency of tea is related to polyphenols, including catechins such as ECG and EGCG, flavonols such as quercetin and myricetin, and tannins such as procyanidin B2. The umami taste of tea is related to theanine, while the bitter taste of tea broth is contributed by alkalines such as theophylline and bitter amino acids such as leucine. Therefore, the influence of temperature and time during the drying stage on these tea flavor substances can be further explored. Metabolic pathway analysis showed that purine metabolism and tyrosine metabolism were important factors for the sensory quality differences in Rizhao green tea at different processing stages. This study helps to understand the metabolic mechanisms of Rizhao green tea at different processing stages and provides a new theoretical basis for the development of green tea flavor.

## Figures and Tables

**Figure 1 molecules-28-07447-f001:**
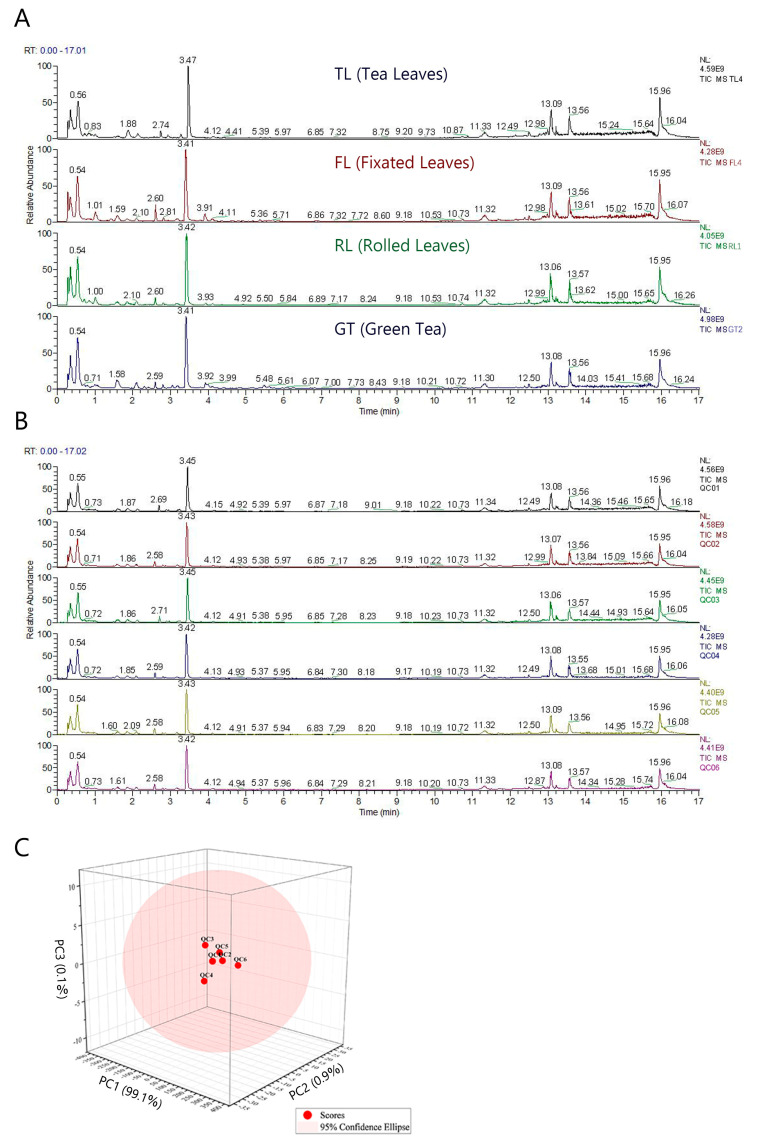
Obtained by C18 column in ESI+ mode: (**A**) TIC of tea samples at different processing stages; (**B**) TIC of QC samples; (**C**) 3D principal component analysis of samples.

**Figure 2 molecules-28-07447-f002:**
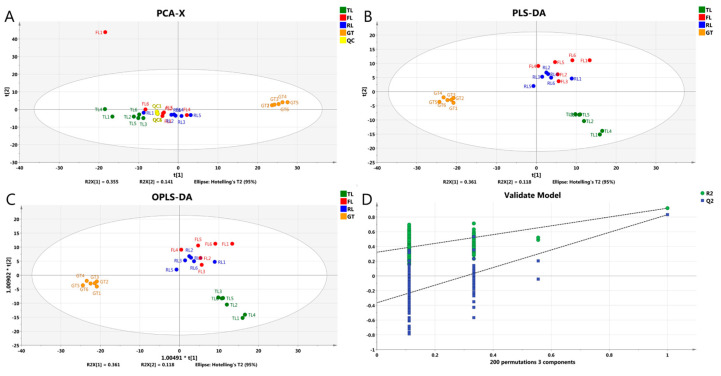
Multivariate statistical analyses of Rizhao green tea at different processing stages. (**A**) PCA score plot; (**B**) PLS-DA score plot; (**C**) OPLS-DA score plot; (**D**) cross-validation plot of the OPLS-DA model with 200 permutation tests.

**Figure 3 molecules-28-07447-f003:**
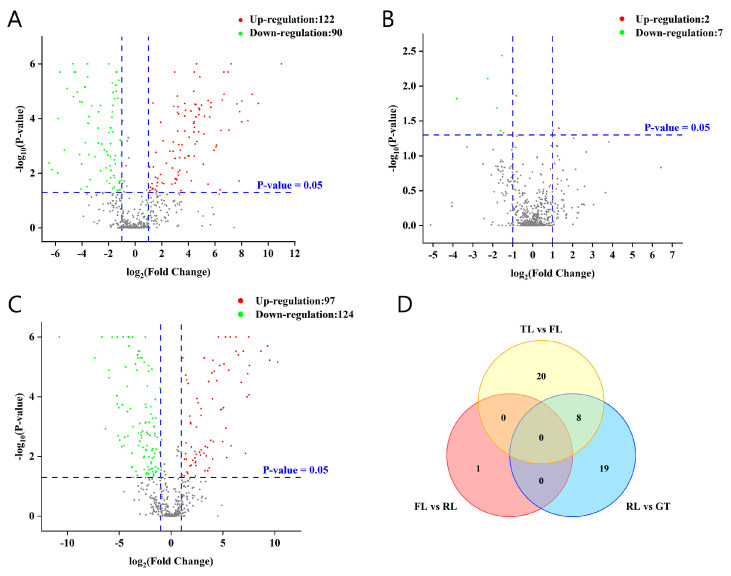
Differential metabolite analysis was performed between Rizhao green tea samples at different processing stages. (**A**) Volcano plots of TLs vs. FLs. (**B**) Volcano plots of FLs vs. RLs. (**C**) Volcano plots of RLs vs. GT. (**D**) Venn diagram of differential metabolites based on the volcano plots and value of VIP.

**Figure 4 molecules-28-07447-f004:**
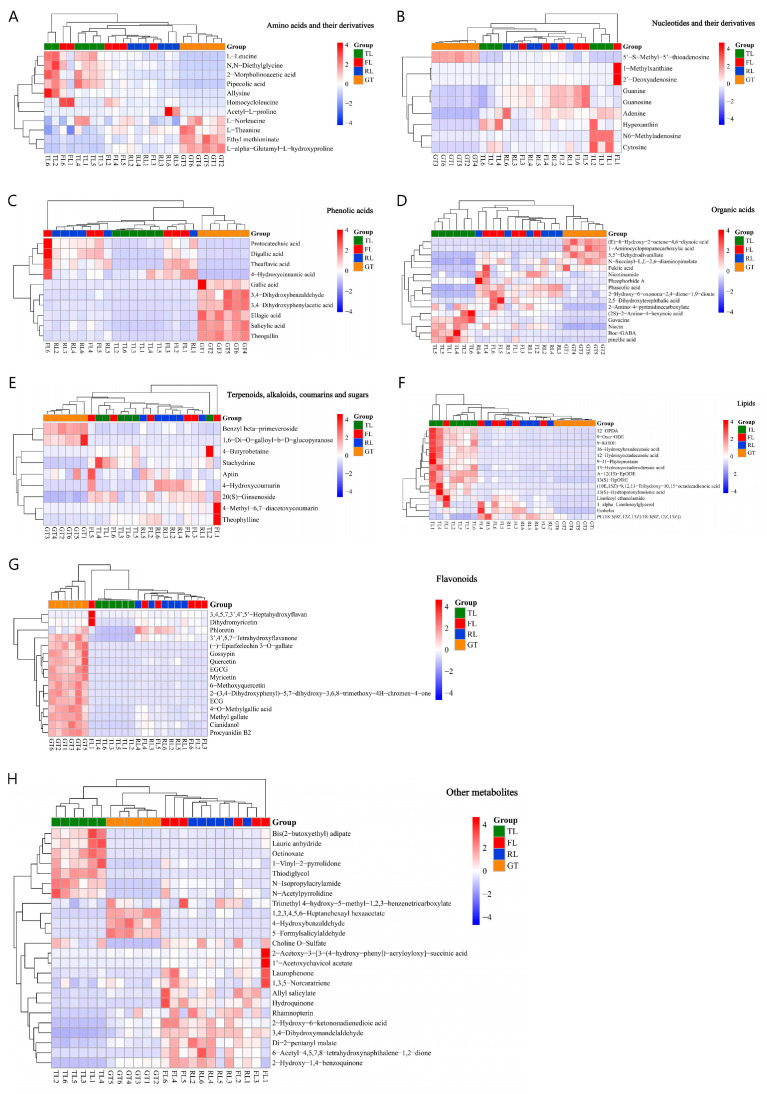
Heat map of the levels of the 112 differential non-volatile metabolites during different processes. (**A**) Amino acids and their derivatives. (**B**) Nucleotides and their derivatives. (**C**) Phenolic acids. (**D**) Organic acids. (**E**) Terpenoids, alkaloids, coumarins, and sugars. (**F**) Lipids. (**G**) Flavonoids. (**H**) Other metabolites.

**Figure 5 molecules-28-07447-f005:**
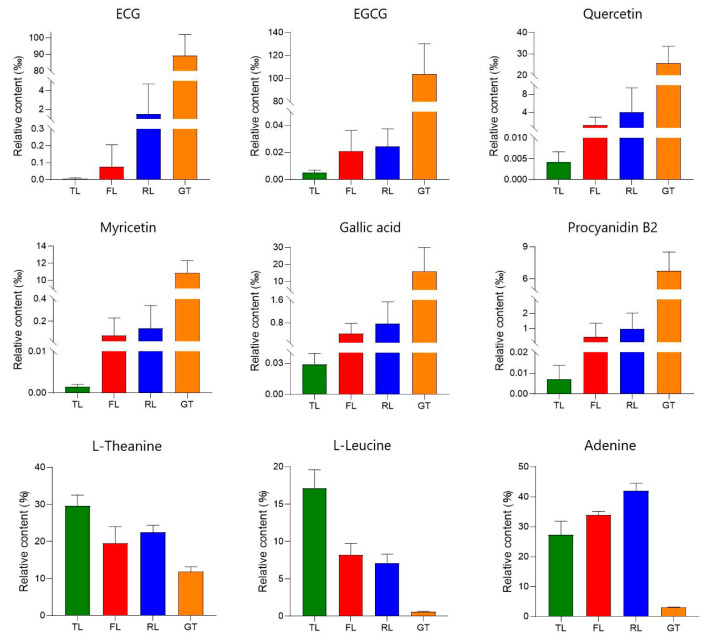
Change trend of compounds related to tea quality characteristics in processes.

**Figure 6 molecules-28-07447-f006:**
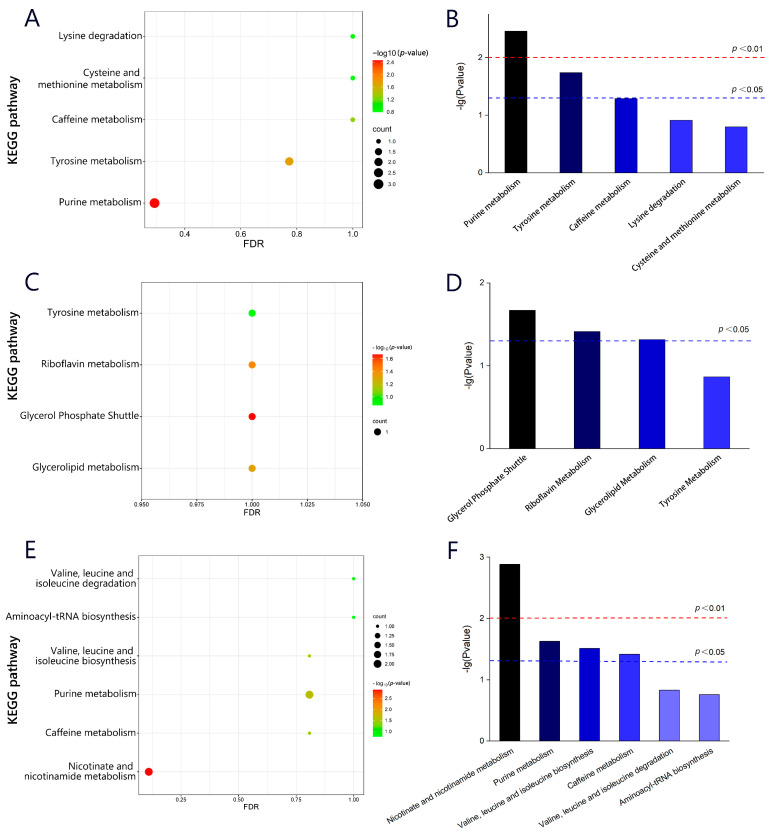
Distribution graph of differential metabolites in different metabolic and biosynthetic pathways.

## Data Availability

The authors confirm that the data supporting the findings of this study are available within the article and its Appendix A.

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
