# Peer review of "Study on the Dynamic Changes in Non-Volatile Metabolites of Rizhao Green Tea Based on Metabolomics"

_molecules, 2023, doi:10.3390/molecules28217447_

Round 1

Reviewer 1 Report

Comments and Suggestions for Authors

This study investigate the dynamic changes of non-volatile metabolites during rizhao green tea processing, there are some merits of this work. However, the content of this study is simple and some issues should be addressed.

1.       Figures 1 and 4 should be clearly presented, it is too blurry.

2.       The major issue is identification. Was identification confirmed by analysis of authentic standards? It is not mentioned in the manuscript, while for some of the significant metabolite standards are available on the market. The ion fragments should be provided for the identified metabolites.

3.       More detail discussion should be added for the result obtained.

4.       There are some grammar errors for the manuscript, please carefully checked it.

5.       Actually, similar studies were previously reported, please emphasis the novelty of the manuscript.

6.       There are too many decimal points in Table 1, please revise it.

7.       The change of catechins should be determined by authentic standards using HPLC.

Comments on the Quality of English Language

 Moderate editing of English language required.

Author Response

Response to Reviewer 1 Comments

Point-by-point response to Comments and Suggestions for Authors

Comments 1: This study investigate the dynamic changes of non-volatile metabolites during rizhao green tea processing, there are some merits of this work. However, the content of this study is simple and some issues should be addressed.

Response 1: Thanks for the reviewer’s comments. According to the reviewer’s suggestion, we have improved the clarity of Figure 1 and Figure 4.

Comments 2: The major issue is identification. Was identification confirmed by analysis of authentic standards? It is not mentioned in the manuscript, while for some of the significant metabolite standards are available on the market. The ion fragments should be provided for the identified metabolites.

Response 2: Thanks to the reviewer’s suggestion. As mentioned in the chapter of “1. introduction”, in order to investigate the changes of Rizhao green tea flavor compounds during processing, the non-targeted metabolomics method based on UHPLC-Q Exactive-MS/MS was used to detect the flavor substances in the four processing stages, with the aim of obtaining as much compound data as possible to comprehensively characterize the flavor components in tea, in order to study the dynamic changes of flavor substances in the processing of Rizhao green tea, and thus, the analysis was carried out by using only the relative quantification, and no absolute quantification was performed using the authentic standards.

In addition, the present study identified the flavor substances that contributed more to the overall taste of Rizhao green tea through the screening of differential metabolites and the DOT method, and absolute quantitative experiments could be conducted for these limited number of known flavor compounds in subsequent studies to investigate the mechanism of their changes during processing.

Comments 3: More detail discussion should be added for the result obtained.

Response 3: Thanks for the reviewer’s comments. The aim of this study was to investigate the trend of non-volatile substances during the processing of Rizhao green tea, to screen differential metabolites by multivariate statistical analysis, calculation of VIP values, and to identify the main contributors to the flavor of Rizhao green tea using the DOT method, which were classified into seven categories, including amino acids, nucleotides, flavonoids, phenolic acids, lipids, organic acids, as well as other metabolites, and had been meticulously discussed in terms of their changing rules, which were ultimately combined with the enrichment analysis of the metabolic pathways. Therefore, considering the length of the manuscript, no additional discussion will be added.

Comments 4: There are some grammar errors for the manuscript, please carefully checked it.

Response 4: Thank you so much for your careful check. We feel sorry for these error and it were rectified at line 45, 95, 130 and so on.

L.47-49: “And in 2020, it was successfully included in the the EU certified China Geographical Indication Protection Catalogue.” was rectified as “And in 2020, it was successfully included in the the EU certified China Geographical Indication Protection Catalogue.”

L.109-111: “In addition, differential metabolites between adjacent processing stages were screened, and their contributions to sample taste were determined using DOT method.” was rectified as “In addition, differential metabolites between adjacent processing stages were screened, and their contributions to sample taste were determined using the DOT method.”

L.172-174: “Figure 2D shows the 200 tmes confidence test, indicating that the results of multivariate statistical analysis are reliable.” was rectified as “Figure 2D shows the 200 times tmes confidence test, indicating that the results of multivariate statistical analysis are reliable.”

L305-L306: “Peng et al. 36 confirmed that L-theanine and flavanols can react to form EPSFs under dry tea conditions, and EPSFs increases during tea baking” was rectified as “Peng et al. 36 confirmed that L-theanine and flavanols can react to form EPSFs under dry tea conditions, and EPSFs increase increases during tea baking”

L.313: “Some nucleotides and their derivatives also have a umami taste.” was rectified as “Some nucleotides and their derivatives also have an a umami taste.”

L.415: “The original data was subjected to peak alignment.” was rectified as “The original data were was subjected to peak alignment.”

L.387-390: “And mixed solution was centrifuged at 12000 rpm for 10 min at 4℃. The supernatant was filtered with a 0.22 μm membrane into labeled autosampler vials and stored at 4℃ for later analysis. Each sample was prepared at least in 6 times.” was rectified as “And the mixed solution was centrifuged at 12000 rpm for 10 min at 4℃. The supernatant was filtered with a 0.22 μm membrane (ANPEL, Shanghai, China) into labeled autosampler vials by disposable syringes (Shinva ANDE, Shandong, China), and stored at 4℃ for later analysis. Each sample was prepared at least in 6 times.”

L.395-L396: “Column temperature were set at 40℃ and 30℃, and injection volume were 4 μL and 3 μL.” was rectified as “The column temperature was were set at 40℃ and 30℃, and the injection volume was were 4 μL and 3 μL.”

Comments 5: Actually, similar studies were previously reported, please emphasis the novelty of the manuscript.

Response 5: Thanks for the reviewer’s comments. With such a unique climatic and geographic position, Rizhao green tea has unique qualities and is differentiated with other green tea. In this study, we investigated the effect of tea manufacturing process on the formation and changes of tea flavor substances of metabolomics using UHPLC-Q Exactive-MS/MS. The study is conducive to the improvement and enhancement of tea flavor and quality. The novelty of the manuscript was reflected in the chapter of “1. introduction”:

L.54-56: “Currently, there is insufficient basic research on the quality and flavor substance changes of Rizhao green tea during the processing.”

L.98-100: “This method has been applied to food fields such as alcohol 15, but has been less applied in the determination of the taste characteristics of Rizhao green tea.”

Comments 6: There are too many decimal points in Table 1, please revise it.

Response 6: Thank you for your comments. The relative content of some substances in tea leaves is low, e.g., myricetin is only 0.0004%. In order to show its relative content more accurately, 4 decimal places were used.

Comments 7: The change of catechins should be determined by authentic standards using HPLC.

Response 7: Thanks for your suggestion. We will follow up with absolute quantification of several contributing prominent substances identified in this study using HPLC in future research.

Reviewer 2 Report

Comments and Suggestions for Authors

The aim of the study was to assess the impact of processing Rizhao green tea leaves on the content of non-volatile compounds. The work used modern analytical methods (UHPLC-Q Exactive MS) and many statistical methods to process the results. The aim of the work is interesting and has practical significance for food producers. The manuscript is written carefully, the results are discussed, the references are appropriate. The work should include information on the leaf preparation method and improve the readability of the drawings. Detailed comments are provided below.

I recommend adding a description of the abbreviations (TL, FL, RL and GT) used to the titles of the figures (figures 4, 5). This should make the figures included in the work very easy to read.

Were the parameters for handling leaves after harvest selected in a preliminary experiment, or were they selected on the basis of materials used in large-scale (industrial) tea production? I recommend explaining the choice of time and temperature for leaf processing.

More information should be provided regarding the preparation of tea leaves. What device was used to heat the samples to 230°C, dry and roll them? How were they performed? Manually or was some device used?

A sonication process was used to extract the samples. The information only concerns the processing time and temperature, but there is no information about the frequency or input power.

Information specifying the type of syringe filter used in UPLC analysis should be written.

The conclusions in the work lack attention to the direction of future research, they take into account the experience. What aspects require further research? Maybe better drying conditions should be developed so that this process affects the quality of the tea as little as possible? This should be indicated in the conclusions.

The results obtained in the research have a very important practical aspect. I recommend adding one sentence to the manuscript abstract, which will be addressed to the food industry, with practical advice for it.

I also recommend increasing the font size in the figures to make them more legible. It is very difficult to read the Y scale in Figure 6 A, C and F and the signatures of all axes in Figure 1C.

Author Response

Summary

Thank you very much for taking your time to review this manuscript 2624124. We really appreciate all your generous comments and suggestions, these comments are very helpful to improve the quality of the manuscript. We have carefully revised our manuscript and words in blue are the change. Now I response the reviewers’ comments with a point by point.

Response to the Reviewers’ comments

Thanks for your generous comments. According to your advice, we amended the relevant part in manuscript. All of your questions were answered one-by-one.

Point-by-point response to Comments and Suggestions for Authors

Comments 1: I recommend adding a description of the abbreviations (TL, FL, RL and GT) used to the titles of the figures (figures 4, 5). This should make the figures included in the work very easy to read.

Response 1: Thanks for the reviewer’s comments. We added the description of the abbreviations (TL, FL, RL and GT) in the section of “3. Materials and methods” (L.377-379):“Tea samples (n=6) from four key processing stages were named as tea leaves (TL), fixated leaves (FL), rolled leaves (RL), and green tea (GT).”

Furthermore, we showed the abbreviations and full names in Figure 1. Therefore not much description has been added to the later figures.

Comments 2: Were the parameters for handling leaves after harvest selected in a preliminary experiment, or were they selected on the basis of materials used in large-scale (industrial) tea production? I recommend explaining the choice of time and temperature for leaf processing.

Response 2: Thanks for the reviewer’s comments. The parameters for handling leaves after harvest were selected on the basis of materials used in large-scale (industrial) tea production. (L.371-375): “The fresh tea leaves with one or two leaves and one bud were repeatedly collected and screened by tea makers to ensure the samples are more representative in appearance, colour, shape, and other aspects, and then converted into commercial premium green tea of first grade, according to the standard process of withering, fixation, rolling, and drying.”

Comments 3: More information should be provided regarding the preparation of tea leaves. What device was used to heat the samples to 230°C, dry and roll them? How were they performed? Manually or was some device used?

Response 3: Thanks for the reviewer’s comments. The Rizhao tea samples in this study were made into commercial premium green tea of first grade using standard processes and devices. The processing time and temperature for the fixation, rolling, and drying stages have been explained in “3. Materials and methods”. (L.371-377): “The fresh tea leaves with one or two leaves and one bud were repeatedly collected and screened by tea makers to ensure the samples are more representative in appearance, colour, shape, and other aspects, and then converted into commercial premium green tea of first grade, according to the standard process of withering, fixation, rolling, and drying. Postharvested samples were spread out from 4 h to 6 h. Following this step, tea samples were fixed at 230℃ for 1 min to reach a relative humidity of 60%, rolled at room temperature for 35 min, and dried at 90℃ for 20 min to produce Rizhao green tea.”

Comments 4: A sonication process was used to extract the samples. The information only concerns the processing time and temperature, but there is no information about the frequency or input power.

Response 4: Thanks for the reviewer’s comments. According to the reviewer’s suggestion, we have further added the details as follows:

L.364-366: “Tea samples were centrifuged by SIGMA 3-18K (Sigma-Aldrich, Saint Louis, Missouri, USA).” was rectified as “Tea samples were ultrasonically extracted by SB-800 (Scientz, Ningbo, China) and centrifuged by SIGMA 3-18K (Sigma-Aldrich, Saint Louis, Missouri, USA).”

L.387-388: “An ultrasonic procedure of 30 min under room temperature was adapted to extract metabolites.” was rectified as “An ultrasonic procedure at 40 KHz for of 30 min under room temperature was adapted to extract metabolites.”

Comments 5: Information specifying the type of syringe filter used in UPLC analysis should be written.

Response 5: Thanks for the reviewer’s comments. According to the reviewer’s suggestion, we have further added the details as follows:

L.389-391: “The supernatant was filtered with a 0.22 μm membrane into labeled autosampler vials and stored at 4℃ for later analysis.” was rectified as “The supernatant was filtered with a 0.22 μm membrane (ANPEL, Shanghai, China) into labeled autosampler vials by disposable syringes (Shinva ANDE, Shandong, China), and stored at 4℃ for later analysis.”

Comments 6: The conclusions in the work lack attention to the direction of future research, they take into account the experience. What aspects require further research? Maybe better drying conditions should be developed so that this process affects the quality of the tea as little as possible? This should be indicated in the conclusions.

Response 6: Thank you for your comments. According to the reviewer’s suggestion, we have further added the details as follows:

L.449-455: “The umami taste of tea is related to theanine, while the bitterness taste of tea broth is contributed by alkaline such as theophylline and bitter amino acids such as leucine. Metabolic pathway analysis showed that purine metabolism and tyrosine metabolism were important factors for the sensory quality differences of Rizhao green tea at different processing stages.” was rectified as “The umami taste of tea is related to theanine, while the bitterness taste of tea broth is contributed by alkaline such as theophylline and bitter amino acids such as leucine. Therefore, the influence of temperature and time during the drying stage on these tea flavor substances can be further explored. Metabolic pathway analysis showed that purine metabolism and tyrosine metabolism were important factors for the sensory quality differences of Rizhao green tea at different processing stages.”

Comments 7: The results obtained in the research have a very important practical aspect. I recommend adding one sentence to the manuscript abstract, which will be addressed to the food industry, with practical advice for it.

Response 7: Thanks for the reviewer’s comments. According to the reviewer’s suggestion, we have further added the details as follows:

L.22-23: “The results of this study provide a theoretical basis for tea processing.” was rectified as “The results of this study provide a theoretical basis for tea processing and practical advice for the food industry.”

Comments 8: I also recommend increasing the font size in the figures to make them more legible. It is very difficult to read the Y scale in Figure 6 A, C and F and the signatures of all axes in Figure 1C.

Response 8:  Thank you for your comments. According to the reviewer’s suggestion, we have improved the clarity of Figure 1C and Figure 6A, C, F, in addition to increasing the font size as follows:

Fig. 1. Obtained by C18 column in ESI+ mode (A) TIC of tea samples with different processing (B) TIC of QC samples (C) 3D-principal component analysis of samples.

Fig. 6. Distribution graph of differential metabolites in different metabolic and biosynthetic pathways.
